# Anisotropic photoemission time delays close to a Fano resonance

Claudio Cirelli[1,2], Carlos Marante[3], Sebastian Heuser[1], C.L.M. Petersson[3], Álvaro Jiménez Galán[3,4],
Luca Argenti[3,5], Shiyang Zhong[6], David Busto [6], Marcus Isinger[6], Saikat Nandi[6], Sylvain Maclot[6],
Linnea Rading[6], Per Johnsson[6], Mathieu Gisselbrecht[6], Matteo Lucchini[1,10], Lukas Gallmann [1],
J. Marcus Dahlström[7], Eva Lindroth [7], Anne L'Huillier[6], Fernando Martín[3,8,9] & Ursula Keller [1]

Electron correlation and multielectron effects are fundamental interactions that govern many physical and chemical processes in atomic, molecular and solid state systems. The process of autoionization, induced by resonant excitation of electrons into discrete states present in the spectral continuum of atomic and molecular targets, is mediated by electron correlation. Here we investigate the attosecond photoemission dynamics in argon in the 20–40 eV spectral range, in the vicinity of the $3s^{-1}np$ autoionizing resonances. We present measurements of the differential photoionization cross section and extract energy and angle-dependent atomic time delays with an attosecond interferometric method. With the support of a theoretical model, we are able to attribute a large part of the measured time delay anisotropy to the presence of autoionizing resonances, which not only distort the phase of the emitted photoelectron wave packet but also introduce an angular dependence.

[1] Physics Department, ETH Zurich, 8093 Zurich, Switzerland. [2] Empa, Swiss Federal Laboratories for Material Science and Technology, Laboratory for Advanced Analytical Technologies, CH-8600 Dübendorf, Switzerland. [3] Departamento de Química, Módulo 13, Universidad Autónoma de Madrid, 28049 Madrid, Spain. [4] Max Born Institute, Max Born Strasse 2ᵃ, D-12489 Berlin, Germany. [5] Department of Physics and CREOL College of Optics & Photonics, University of Central Florida, Orlando, FL 32816, USA. [6] Department of Physics, Lund University, SE-221 00 Lund, Sweden. [7] Department of Physics, Stockholm University, AlbaNova University Center, SE-10691 Stockholm, Sweden. [8] Instituto Madrileño de Estudios Avanzados en Nanociencia (IMDEA-Nano), Cantoblanco, 28049 Madrid, Spain. [9] Condensed Matter Physics Center (IFIMAC), Universidad Autónoma de Madrid, 28049 Madrid, Spain. [10]Present address: Department of Physics, Politecnico di Milano, Piazza L. da Vinci 32, 20133 Milano, Italy. Correspondence and requests for materials should be addressed to C.C. (email: claudio.cirelli@psi.ch)

The development of attosecond sources based on high-order harmonic generation (HHG) in gases has opened the possibility to investigate electron dynamics on its natural timescale[1]. Recently, pump-probe experiments with the pump in the extreme ultraviolet (XUV) range and the probe in the infrared (IR)[2,3] showed the feasibility to access and measure photoemission delays, which were introduced theoretically in the 50s[4,5] and reviewed in[6]. Since then, it has become a very active field of research in attosecond science[7,8]. Attosecond photoemission dynamics was studied experimentally with the attosecond streak camera in atomic[2,9] and solid state targets[10], and addressed theoretically[2,11,12]. This was complemented by detailed interferometric measurements using the RABBIT (reconstruction of attosecond beatings by interference of two-photon transitions) technique[3] in atoms[13–16], molecules[17] and solid targets[18].

In the simplest case, when the electron is promoted into a flat (non-resonant) continuum by direct laser-assisted photoionization, the measured delay after absorbing a single XUV photon is related to the phase shifts of the departing electron induced by the ionic potential and laser field, respectively. One part of this so-called atomic time delay is the Wigner delay[4,5], which can be expressed as the energy derivative of the scattering phase and is equivalent to the group delay of the departing electron wave packet. Thus the Wigner delay has a direct link to the classical trajectory with the center of the electron wave packet following the Ehrenfest's theorem[19]. In absence of resonances, this quantity can be accessed with attosecond techniques if other contributions are carefully subtracted[11,20].

Compared to the direct ionization into the continuum, the situation becomes more complicated when ionization occurs in the vicinity of autoionizing states[9,21,22]. Autoionizing states are highly excited short-living states that ultimately decay into the continuum, thus opening an alternative ionization path. The interference between the direct and autoionizing pathways gives rise to the well-known Fano profiles in the photoionization cross sections[23], which have been directly measured in atoms[24] and molecules[25] with high precision using for instance monochromatic synchrotron radiation. Despite their value, these studies are unable to access the correlated dynamics of the photoionized resonant electron wave packet. This information can be obtained by measuring atomic time delays with attosecond techniques[17,26]. As an example, the build-up of a Fano resonance in time domain has been observed using both RABBIT[27] and attosecond transient absorption[28]. Yet, to date the angular behavior of resonant atomic time delays, from which we can gain more insight, remains unexplored.

Recently, techniques combining multicolor fields (e.g., XUV-IR) with electron momentum detection have been used to retrieve angular-resolved phase and amplitude of ionizing electron wave packets[29,30]. A benchmark study performed with helium[31] in non-resonant conditions showed that photoionization time delays had almost no angular dependence, except at large angles relative to the laser polarization. For He, the absorption of a single XUV photon opens only one ionization channel ($1s \rightarrow \varepsilon p$), but a second IR probe photon is required in the attosecond RABBIT technique to measure time delays. Therefore, the observed time delay anisotropy was attributed to phase differences between final quantum states with different angular symmetry resulting from two-photon (XUV+IR) ionization.

In this work, we demonstrate how the angular dependence of the atomic time delays is affected by correlation effects associated to the mechanism of autoionization, thus giving access to angle-resolved multi-electron dynamics on the attosecond time scale. We present measurements of the differential photoionization cross section of argon in a spectral energy range where the $3s^1 3p^6 np$ series of autoionizing resonances can be efficiently

populated[32,33]. The measured photoelectron angular distributions (PADs) obtained by one-photon absorption are in excellent agreement with static measurements from studies performed with synchrotron radiation. In addition, we get access to the angle-resolved atomic time delays at photon energies spanning an autoionizing resonance. Our results show that the atomic time delay measured near these resonances depends strongly on the electron emission angle relative to the polarization of the ionizing XUV field. The ratio of the ionization channels $3p \rightarrow \varepsilon d$ and $3p \rightarrow \varepsilon s$ abruptly changes across the resonances, leading to a strong variation of the ionization delay with the electron emission angle and energy.

## Results

**Experimental results.** Results from two experiments, performed at ETH Zurich and Lund University are presented. Figure 1 shows the HHG spectra used in these experiments. In both cases, the 17th harmonic is resonant with an autoionizing state, the $3s^{-1}5p$ (ETH) and the $3s^{-1}4p$ (Lund) states[34]. The two experiments are both based on the RABBIT technique, but using different detection setups presented in details in the Methods section (see also refs. 26 and 35).

The atomic delays were measured with the XUV-IR interferometric RABBIT technique[36,37]. When an atom with ionization potential $I_p$ is ionized by an XUV attosecond pulse train (APT), photoelectrons are released in the continuum at discrete kinetic energies equal to $E_{kin} = E_{HH} - I_p$, where $E_{HH} = (2q+1)\hbar\omega$ defines the XUV photon energy comb of the APT ($\omega$ is the IR laser frequency). When an IR dressing field is added, we obtain two-color two-photon transitions with a photoelectron spectrum that exhibits additional sidebands (SBs) at energies in-between two consecutive APT comb peaks[38]. These energies correspond to the absorption of an XUV photon combined with the additional absorption or emission of an IR photon. Any SB energy can be reached by two different interfering ionization channels[37]. As explained in more details in the discussion below, the amplitude

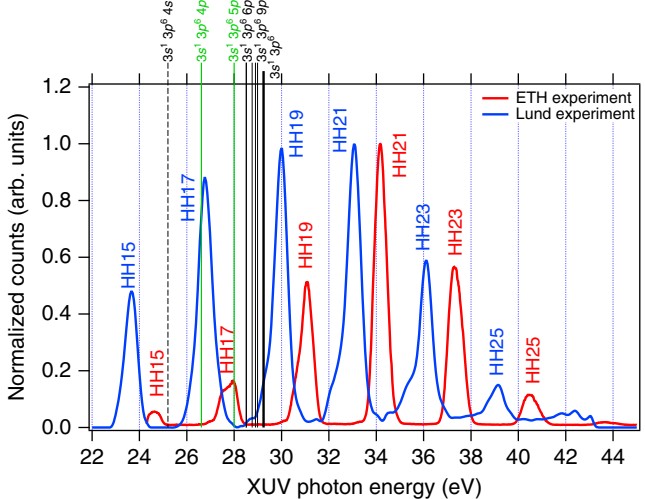

**Fig. 1** Extreme ultraviolet spectra of the attosecond pulse trains used in the experiment. The XUV radiation is generated by focusing the IR beam into an argon target. The vertical lines show the energy position of the $3s^{-1}np$ series of autoionizing states converging to the $3s$ threshold. In the ETH experiment (red line), the $5p$ state (at 27.99 eV, highlighted in green) is resonant with harmonic 17 (HH17), while in the Lund experiment (blue line) it is the $4p$ (26.6 eV, also highlighted in green). The black dashed line indicates the position of the $3s^{-1}4s$ autoionizing state

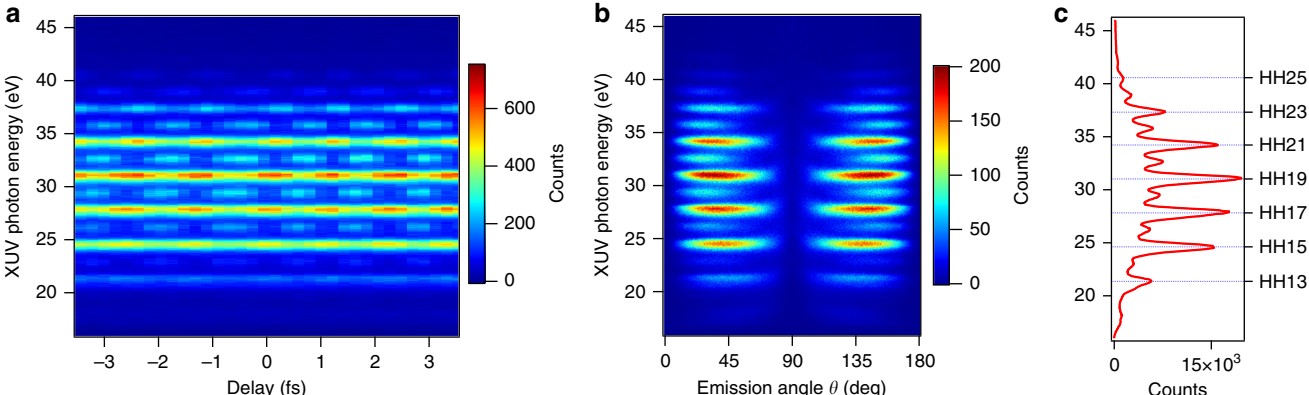

**Fig. 2** RABBIT measurements obtained with the ETH experimental setup. **a** Angle-integrated photoelectron spectrum as a function of the XUV-IR delay. **b** Delay-integrated photoelectron spectrum as a function of the emission angle $\theta$ relative to the common axis of polarization of the XUV and IR pulses. **c** Integration of the spectrogram in angle and delay results in the 1D spectrum where sidebands appear between consecutive harmonics

of the sideband signal oscillates as:

$$A_{SB} \propto \cos(2\omega\tau - \Delta\phi_{atto} - \Delta\phi_{atomic}), \qquad (1)$$

where $\Delta\phi_{atto}$ is the phase difference between the consecutive frequency comb peaks of the APT and therefore corresponds to the so-called attochirp[39], while $\Delta\phi_{atomic}$ is the accumulated atomic phase difference between the two quantum paths (absorption and emission of an IR photon)[36].

Our experimental setups allow us to record angle-resolved RABBIT spectrograms. Fig. 2 shows examples of the results obtained with the ETH setup. Similar data were measured in the Lund experiment. Fig. 2a presents an angle-integrated spectrogram, displaying clear sideband oscillations. Fig. 2b shows a delay-integrated photoelectron spectrum as a function of the emission angle $\theta$, defined relative to the common XUV and IR polarization axis. We determine the photoelectron angular distributions for several discrete photon energy values, over all emission angles except in the region $\theta = 0$, where the detection efficiency of the reaction microscope drops. Fig. 2c presents the angle- and delay-integrated spectrum.

A cosine fit of the (angular-resolved) sideband signal as a function of the delay between XUV and IR pulses returns the value of the total phase $\Delta\phi = \Delta\phi_{atto} + \Delta\phi_{atomic}$[3]. By subtracting $\Delta\phi_{atto}$, the energy- and angle-dependent atomic time delay, defined as $\tau_{atomic} \approx \Delta\phi_{atomic}/2\omega$[18,40], can be retrieved. The atomic time delay can in turn often be written as a sum of two contributions: $\tau_{atomic} \approx \tau_W + \tau_{cc}$[11]. The first term is related to one-photon ionization and, in the case of single-channel photoioniza-tion, is the Wigner delay ($\tau_W \approx \Delta\eta_\ell/2\omega$), where $\eta_\ell$ is the scattering phase and $\ell$ the angular momentum. The second term arises from laser-induced continuum-continuum transitions $\tau_{cc} \approx \Delta\varphi_{cc}/2\omega$.

**Delay-integrated asymmetry parameters.** In spherical coordinates, the photoelectron angular distribution $d\sigma/d\Omega$ measured within a solid angle $d\Omega = \sin\theta d\theta d\varphi$ and resulting from the (multiphoton) photoionization of atoms by linearly polarized photons is given by[29]:

$$\frac{d\sigma}{d\Omega} = \frac{\sigma}{4\pi}\left[1 + \sum_{j=1}^{2L_{max}} \beta_j P_j(\cos\theta)\right], \qquad (2)$$

where $\sigma$ is the total photoionization cross section, $\theta$ is the angle between the emitted photoelectron and the polarization axis of the XUV light, $L_{max}$ is the maximum electron's angular momentum to which the expansion in terms of the Legendre

polynomials $P_j$ of order $j$ is performed. The $\beta_j$ parameters are the coefficients of the Legendre polynomials. The photoelectron angular distributions are symmetric and we expect the odd ($\beta_1$, $\beta_3$ and so on) order parameters to be zero.

In a first analysis, we extract the energy-dependent values of the anisotropy parameters $\beta$ from the PADs. It is well-known that autoionization leads to a change of the anisotropy parameter $\beta$[32,41,42].

Figure 3a, b shows the experimental photoelectron angular distributions in polar coordinates (panels a and b) for harmonic order 17 (HH17) and sideband 16 (SB16), using the harmonic spectrum shown in red in Fig. 1 (ETH). The PADs are constructed by filtering the counts of the delay-integrated spectrogram (Fig. 2b) with a 0.7 eV wide energy window centered at the harmonic (or sideband) peak. The green lines represent the fit of the distributions, which account for the detector geometry (Eq. (2) is multiplied by $\sin(\theta)$ to account for the geometrical effect related to the solid angle]. In the case of single-photon absorption (harmonics), the sum in Eq. (2) includes only one term, $j = 2$, and the fit of the distributions returns the values of $\beta_2$. For two-photon absorption (sidebands), we stopped the expansion of Eq. (2) at $j = 4$. The fit of the distributions returns vanishing $\beta_1$ and $\beta_3$, which is consistent with the fact that our PADs are left-right symmetric.

Figure 3c, d show the coefficients of the Legendre polynomials (Eq. 1) as a function of the absorbed photon energy for both the harmonic (c, $\beta_2$ only) and sideband peaks (d, $\beta_2$ and $\beta_4$). Our data, indicated in red (ETH) and blue (Lund) compare well with synchrotron measurements (black dots)[33], thus validating our experiments. The $3s^{-1}4p$ state clearly influences the $\beta_2$ coefficient at the 17th harmonic and 16th sideband (SB16) energies in the Lund experiment, while the effect of the $3s^{-1}5p$ state is weakly observed on SB16 in the ETH results. The variation of the $\beta_2$ coefficient for SB16, larger than that observed in HH17, (Lund experiment) might be influenced by the presence of the $3s^{-1}4s$ autoionizing state (25.2 eV, see Fig. 1), which can be populated by two-photon transitions with the 15th or 17th harmonics. Note that the large bandwidth of the XUV and IR radiation used in the present work significantly blurs the effect of the resonances compared to synchrotron radiation. The different widths of the two resonances, 80 meV for the $4p$ and 28.5 meV for the $5p$[32] explain the stronger effect observed in the Lund experiment.

**Delay-dependent asymmetry parameters.** In comparison to static data acquired at synchrotrons, our pump-probe measurements show worse energy resolution, however they provide access

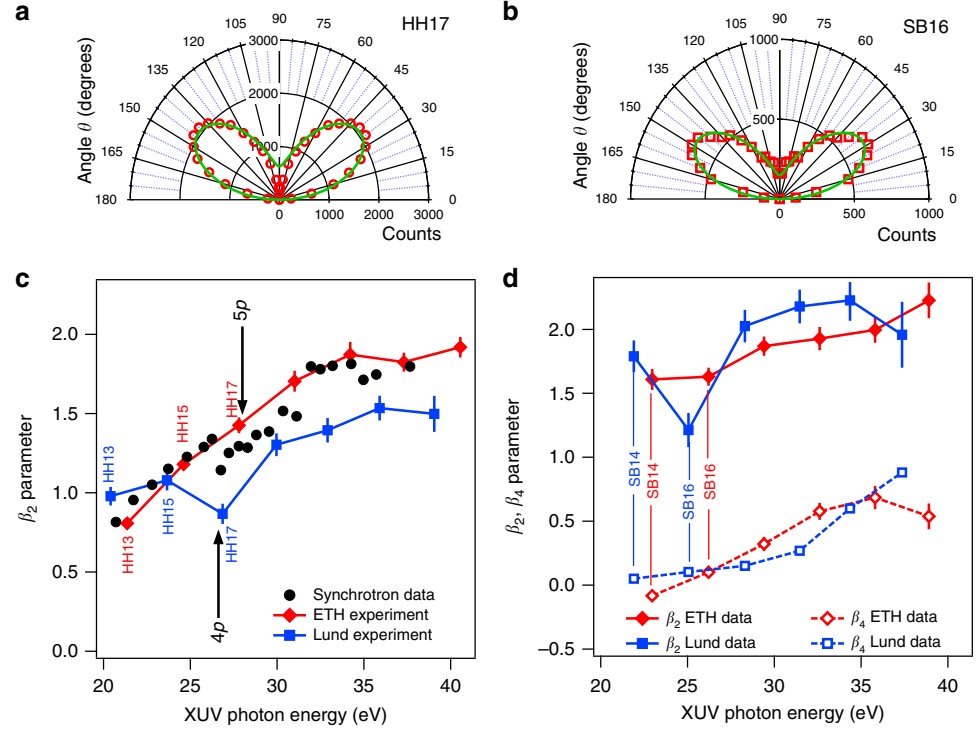

**Fig. 3** Photoelectron angular distributions and $\beta$ parameters. **a** and **b** represent two photoelectron angular distributions (PAD) in polar coordinates for electron kinetic energies corresponding to HH17 and SB16, respectively, as measured in the ETH experiment with the reaction microscope detector. The green solid lines are the fit of Eq. 1, multiplied by $\sin(\theta)$ to account for the detector geometry, to the data. **c** Values of $\beta_2$ parameter as a function of photon energy sampled at the harmonic (solid line) obtained in the ETH (in red) and Lund (in blue) experiments. The black dots are taken from[33] and the arrows at 26.6 eV and 28 eV indicate the positions of the $3s\rightarrow4p$ and the $3s\rightarrow5p$ autoionization resonances. **d** Values of $\beta_2$ parameter (solid lines) and $\beta_4$ parameter (dashed lines) as a function of photon energy sampled at sideband energy positions for the ETH (in red) and Lund (in blue) experiments. In panels **c** and **d**, the data represent the mean value extracted by independent datasets, while the error bars indicate the standard deviation

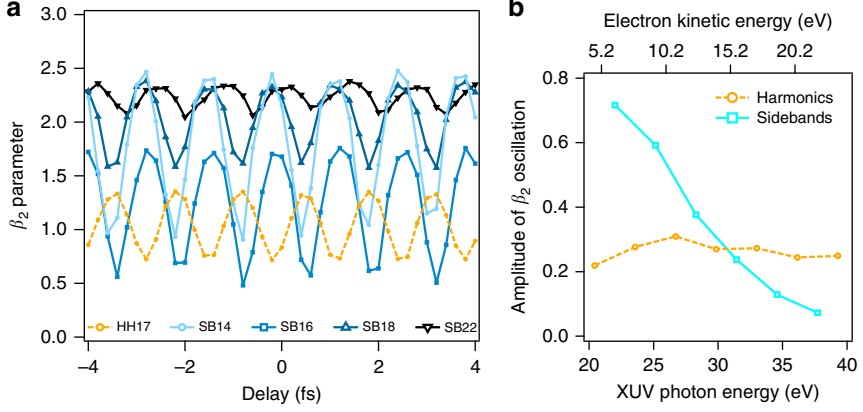

**Fig. 4** Time-dependent $\beta_2$ parameters. Panel **a** shows the values of $\beta_2$ parameter for SB14, SB16, SB18 (blue symbols), SB22 (black symbols) and HH17 (yellow symbols), extracted from a fit of the momentum distributions at different XUV-IR delays in the Lund experiment, for which HH17 is resonant with the $3s^{-1}4p$ autoionizing state. **b** Amplitude of $\beta_2$ oscillations as a function of kinetic energy. For the harmonics, it is approximately constant, while it decreases for the sidebands

to complementary time-dependent information. Fig. 4a shows the behavior of $\beta_2$ as a function of XUV-IR delay for harmonics and sidebands. Here, we present the data from the Lund experiment (obtained with the blue spectrum in Fig. 1). Similar behavior was observed in the ETH experiment. The first observation is that the $\beta_2$ values for the harmonics and sidebands are oscillating with the same frequency $2\omega$, as expected in the RABBIT setting, but with opposite phases. For the sidebands, the variation of $\beta_2$ decreases in amplitude as the photon energy increases, while that of the

harmonics remains constant for the harmonics, as shown in Fig. 4b.

**Angular dependence of the atomic delays**. In Fig. 5 we compare the angular dependence of the atomic time delay for two different sidebands, SB14, which is not affected by the $3s^{-1}np$ resonances (Fig. 5a) and SB16, such that H17 is resonant with the $3s^{-1}5p$ autoionizing state (Fig. 5b). As described in ref. 31, the angle-dependent atomic delay is retrieved by filtering the detected

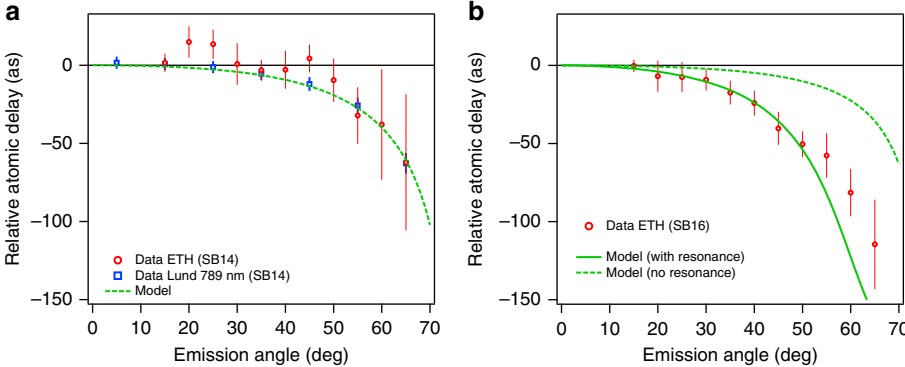

**Fig. 5** Angular-resolved time delays. **a**, **b** show the atomic time delay (red symbols) as a function of electron emission angle for SB14 and SB16 (ETH experiment). Data obtained in Lund for SB14 are also indicated (blue open squares). The delays are referenced to the value retrieved for electrons departing within an opening angle of up to 30 degrees. The green lines show the calculated delays in resonant (solid) and nonresonant (dashed) conditions. The error bars indicate the standard deviation as extracted by a series of independent measurements

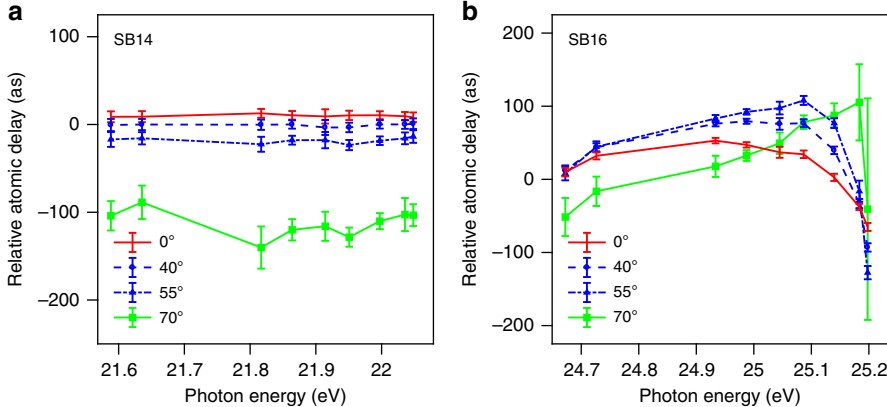

**Fig. 6** Energy and angle-resolved time delays measured in the Lund experiment. Relative atomic delay as a function of sideband photon energy (for two-photon transitions) for different emission angles for SB14 (**a**) and SB16 (**b**). The error bars represent the standard deviation as extracted by a series of independent measurements

photoelectrons at different emission angles with respect to the common polarization axis of the XUV/IR pulses. For each angular sector, a RABBIT spectrogram is constructed and the SB signal is obtained by integrating the spectrogram in an energy window centered at the peak of the SB position. The atomic delays shown in Fig. 5 for each angular sector are referenced against the values retrieved for a sector between 0 to 30 degrees. We have chosen the reference angular range as large as 30 degrees in order to minimize the error in the reference phase. We also present results of a numerical calculation, where the angle-resolved atomic phase is extracted by computing the partial complex amplitudes of the two-photon transition matrix elements. The latter are calculated following two different approaches depending on whether the contribution from a resonant state is included or not. While in the non-resonant case, the matrix elements are evaluated by using second-order time dependent perturbation theory[43], here without including resonant configurations, in the resonant case they include both resonant and non-resonant paths by using a generalization of the Fano configuration interaction formalism for two-photon transitions[44,45]. Further details on how the atomic phase and the matrix elements are calculated are presented in Supplementary Note 1.

The general trend of the angle variation of the atomic time delays is the same as in the case of helium[31], with the delay becoming more and more negative as the emission angle becomes

larger (>50°). For SB14, the experimental data is reproduced by second-order time dependent perturbation theory[43], as can be seen in Fig. 5a. There is no need to include any autoionizing state, because the harmonics involved (HH13 and HH15) are both placed at energies smaller than the first state of the $np$ series ($4p$) and thus are not hitting any resonance (Fig. 1).

For SB16, however, the situation is very different. As seen in Fig. 5b, quantitative agreement between the data and the model is achieved only if the $3s^{-1}5p$ resonance is accounted for. This is accomplished with the theoretical model that was previously validated in helium by comparison with ab-initio calculations[44,45] and extended here to obtain angular information in argon (see Supplementary Note 1). The parameters for the $3s^{-1}5p$ resonance, like the energy position, autoionization width and Fano's $q$ parameter are taken from ab initio calculations based on a multiconfigurational Hartree-Fock (MCHF) approach[46]. When the resonance is neglected, the estimated time delay anisotropy is too small, as indicated by the dashed line.

**Angular-resolved energy dependence of the atomic delays.** Figure 6 presents a spectrally and angularly resolved analysis of the time delays. We here concentrate on the Lund experiment, where we scanned the laser wavelength between 780 and 794 nm so that the 17th harmonic spanned the resonance. In this case, we did not use the normalization applied in Fig. 5, but normalized the delays, for each wavelength, with respect to a line between the

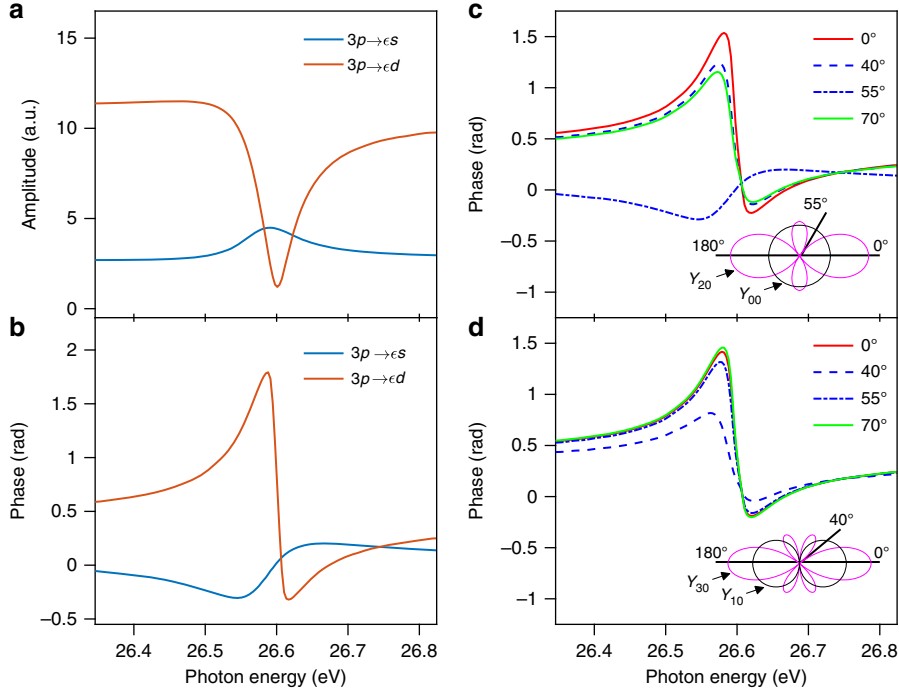

**Fig. 7** Amplitude and phases for one and two-photon transitions. Amplitude (in atomic units) (**a**) and phase (**b**) of one-photon ionization for the $3p \to \varepsilon s$ (blue) and $3p \to \varepsilon d$ channels close to the $3s^{-1}4p$ autoionizing state[46]. Phase variation as a function of energy for one-photon (**c**) and two-photon (**d**) ionization, calculated according to Eq. (7). Note that the *x* axes in this figure refers to the photon energy of the 17th harmonic (single photon transition), whose position is spanned across the 4p resonance. The spherical harmonics involved are indicated as an inset

angle-integrated delays obtained by analyzing SB14 and SB22, similarly to the procedure used in ref. 26. Fig. 6 shows striking differences between the spectral dependence of the delay within SB14 and SB16. For SB14 (Fig. 6a), the delay is relatively flat and decreases for large angles; it barely depends on energy. For SB16, when H17 is resonant with $3s^{-1}4p$, (Fig. 6b), the energy variation of the delay shows the characteristic behavior of the phase change across a Fano resonance[46]. The angle-integrated variation of the delay (not shown here) is lower than that observed in ref. 26, due to the larger XUV and IR bandwidths used in the present work. One of the key conclusions of this work is that the energy variation of the delay also changes with angle. Interestingly, in contrast to the non-resonant case, across the resonance the delay first increases and then decreases with angle. The delay curves at different angles seem to cross at the same point at 25.2 eV. The general behavior of the delay as a function of angle and energy is discussed in the next section.

## Discussion

Here we provide a qualitative explanation of the different effects observed in the experiments, based upon simple arguments. The angle-dependent delay measured in two-photon experiments is theoretically obtained by averaging the angle-resolved RABBIT probability over the orientation of the parent ion (in the case of argon, $m = -1, 0, 1$). However, to understand how the angular dependence of the RABBIT phase arises, it is enough to consider only one orientation. We thus restrict our discussion to $m = 0$, and first describe resonant one-photon ionization, and subsequently anisotropy and delay measurements by the RABBIT technique.

We concentrate here on the $3s^{-1}4p$ autoionizing state, whose effect on the amplitude and phase of the partial ionization channels $3p \to \varepsilon s$ and $3p \to \varepsilon d$ is presented in Fig. 7a, b. The shown data have been obtained from a multiconfiguration Hartree-Fock (MCHF) calculation[46]. Away from the resonant state, the $3p \to \varepsilon d$

channel dominates by a factor ~5 (in amplitude) over the $3p \to \varepsilon s$ channel. However, the amplitude of the $3p \to \varepsilon d$ channel decreases rapidly close to the Fano resonance, while that of the $3p \to \varepsilon s$ channel increases. This obviously affects the geometrical properties of the emitted electron wave packet. The probability amplitude for one-photon ionization from the ground state $3p$, $m = 0$ can be written as:

$$M^{(1)} \propto Y_{20}(\theta)A_2^{(1)}e^{i\eta_2} - Y_{00}(\theta)A_0^{(1)}e^{i\eta_0}, \tag{3}$$

where $A_\ell^{(1)}$ are the photoionization amplitudes for the $3p \to \varepsilon \ell$ channel, $\eta_\ell$ the scattering phases and $Y_{\ell m}(\theta)$ ($m = 0$) the spherical harmonics. Using the amplitudes $A_\ell^{(1)}$ and phases $\eta_\ell$ from ref. 46 (Fig. 7a, b), the variation of the phase of $M^{(1)}$ with energy and angle is shown in Fig. 7c. At small and large angles, the phase variation with energy resembles the phase variation of the $3p \to \varepsilon d$ channel. However, at 54.7° (also called the magic angle), when $Y_{20}$ goes to zero (see inset in Fig. 7), the phase variation is that of the $3p \to \varepsilon s$ channel. In Fig. 7c, the curves cross at around the XUV photon energy of 26.6 eV, when the amplitude of the $3p \to \varepsilon d$ channel becomes small at the resonance. In this case, and in general when there is only one channel for ionization, the phase variation does not depend on emission angle.

The RABBIT technique allows the determination of atomic delays via interferometry. However, as explained below, since it is based upon two-photon XUV-IR transitions, it also changes the angular momentum of the final states and consequently the photoelectron angular distributions.[38,47] For any ionization channel leading to the same final state, our experimental measurement involves three two-photon pathways: $3p \to \lambda \to \ell$, with $(\lambda, \ell) = (0,1), (2,1), (2,3)$. Let us denote the amplitude of the respective pathways by $E_{\ell\lambda}$, the phase by $\varphi_{\ell\lambda}^{(E)}$ for the emission path, where an XUV photon is absorbed and an IR photon is emitted. The angular part is a spherical harmonic $Y_{\ell 0}$. We assume that the XUV and IR fields are delayed by $\tau$ and omit any

additional phase due to the fields (e.g. the attochirp). The two-photon transition amplitude ($m = 0$), can then be written as:

$$M_E \propto e^{-i\omega\tau}\left\{Y_{30}(\theta)e^{i\varphi_{32}^{(E)}}E_{32} + Y_{10}(\theta)\left[e^{i\varphi_{12}^{(E)}}E_{12} - e^{i\varphi_{10}^{(E)}}E_{10}\right]\right\}. \tag{4}$$

Denoting the amplitude and phase of the term within the curly brackets $E(\theta)$ and $\varphi_E(\theta) = \text{Arg}[E(\theta)]$ respectively, and the corresponding quantities for path A $A(\theta)$ and $\varphi_A(\theta)$ , where both photons are absorbed, the RABBIT amplitude can be expressed as

$$M_{SB} \propto e^{-i\omega\tau + i\varphi_E(\theta)}|E(\theta)| + e^{i\omega\tau + i\varphi_A(\theta)}|A(\theta)|. \tag{5}$$

Thus, the sideband intensity simply reads as

$$|M_{SB}|^2 \propto |A(\theta)|^2 + |E(\theta)|^2 + 2|A(\theta)E(\theta)|\cos[2\omega\tau + \varphi_A(\theta) - \varphi_E(\theta)]. \tag{6}$$

As can be seen, in this general case, the phase of the oscillations depends on the emission angle[48], and the angular distributions vary with the delay.

We now examine how Eq. (6) simplifies for particular cases. In the non-resonant case and at high kinetic energy, the two interfering paths become comparable, i.e., $A(\theta) \approx E(\theta)$, $\varphi_A(\theta) \approx \varphi_E(\theta) + \Delta\varphi$, where $\Delta\varphi$ is assumed to be angle-independent, Eq. (6) reduces to

$$|M_{SB}|^2 \propto 2|A(\theta)|^2[1 + \cos(2\omega\tau - \Delta\varphi)] \tag{7}$$

so that the angular distributions have the same form, and hence the same value of asymmetry parameters $\beta$, for all $\tau$. As this approximation should improve with increasing kinetic energy, the PADs should become increasingly independent of the delay, in agreement with the experimental results shown in Fig. 4. Alternatively, as suggested in ref. 49, this behavior might also result from changes in the phases involved in the two-photon transitions as the photon energy increases.

Under the assumption that the resonance does not appreciably affect continuum-continuum transitions, the phase for the channel $3p \rightarrow \lambda \rightarrow \ell$ (e.g. emission path) is approximately equal to $\eta_\lambda^E + \varphi_{cc}^E + \lambda\pi/2$[11], where $\eta_\lambda^E$ is the scattering phase of the intermediate continuum state, and where $\varphi_{cc}^E$ originates from the IR-induced continuum-continuum transition. At high kinetic energy, $\varphi_{cc}^E$ can be calculated with an asymptotic approximation, and is then independent of the intermediate or final state angular momenta[11]. If in addition the $3p \rightarrow s$ channel can be neglected, the phase terms in Eq. (6) factorize out of the curly bracket and, using $\varphi_{cc}^E \approx -\varphi_{cc}^A$ as well as $\Delta\eta_2 = \eta_2^E - \eta_2^A$, Eq (6) becomes

$$|M_{SB}|^2 \propto |A(\theta)|^2 + |E(\theta)|^2 + 2|A(\theta)E(\theta)|\cos\left[2\omega\tau - \Delta\eta_2 - 2\varphi_{cc}^E\right]. \tag{8}$$

In this case, the atomic delay is isotropic. As can be seen in Fig. 5a, this is indeed the case for emission angles $\theta < 45°$. The angular dependence observed for large angles is therefore due to the non-negligible contribution of the $3p \rightarrow s$ channel and/or the breakdown of the asymptotic approximation for the $\varphi_{cc}$ phase. When only one intermediate channel is accessible, as in helium at low photon energies[31], the anisotropy of the atomic delay is solely due the breakdown of the asymptotic approximation. More details on this point are given in the Supplementary Note 2.

We now consider the resonant case. The measured angular dependence of the phase in our two-photon measurement is not a direct "copy" of that for one-photon ionization because the continuum-continuum transition, needed for the phase measurement, projects the resonant state onto different angular momentum states, and thus modifies the angular distribution. To

qualitatively describe the effect of the IR-induced transitions on the one-photon phase, we focus on the $3s^{-1}4p$ resonance and assume that, for the non-resonant XUV + IR absorption path that involves HH15, the one-photon channel $3p \rightarrow s$ is negligible in comparison with the $3p \rightarrow d$ one. This is a reasonable approximation since the latter channel is dominant in this range of photon energies (Fig. 7a). We also neglect the angular momentum dependence of $\varphi_{cc}$. In Eq. (6), $\eta_A(\theta) = \eta_2^A + \varphi_{cc}^A$, which does not depend on angle, thus providing a true reference phase. We also assume that the radial part of $E_{\ell\lambda}$ is proportional to the corresponding one-photon amplitude. The angle-resolved determination of the phase of the RABBIT oscillations as a function of energy allows us to study the variation of the phase of $\eta_E(\theta)$. The results of this simplified model are shown in Fig. 7d. The atomic delay varies with energy similarly to that of the one-photon amplitude at angles where the $3p \rightarrow \varepsilon d$ channel dominates. This behavior changes at the angle 40°, when the spherical harmonic $Y_{30}$ goes to zero. For this angle, the energy dependence of the delay is not that of the $3p \rightarrow \varepsilon s$ channel, as in the one-photon ionization case, but results from the combination of the two channels appearing in the factor that goes with $Y_{10}$ in Eq. (4). As in the one-photon case, the curves cross at the one-photon energy of 26.6 eV, i.e., there is no angular dependence, because the $3p \rightarrow \varepsilon d$ channel becomes negligible. Although this simple model does not allow us to explain all aspects of the experimental results, it does show that it is the delicate interplay between the different open channels that is actually responsible for the complex angular variation of the spectrally resolved phase observed in the experiment (Fig. 6b).

In conclusion, we have presented measurements of the two-color (i.e., XUV- IR) differential photoionization cross section of argon and extracted time-dependent anisotropy parameters as well as energy and angle-dependent atomic time delays. The spectrum of the employed XUV radiation lies in the energy region where several singly excited bound states decay via autoionization. The presence of autoionizing states clearly manifests in the measured time delays for both the narrow $3s^{-1}5p$ and the broad $3s^{-1}4p$ resonances. They are also very visible in the anisotropy parameters extracted from time-integrated photoelectron angular distributions generated by two-photon absorption. These results demonstrate not only that the phase of the photoelectron wave packet is significantly distorted in the presence of resonances, which prevents one from interpreting the Wigner delay as photoemission time delay[9,50], but also that this distortion depends on the electron emission angle. The effect of the resonance on the angular dependence of the atomic delay is due to the existence of several open channels with different angular emission properties and with a varying amplitude across the resonance.

## Methods

**Experimental setup**. In both ETH and Lund experiments, a Ti:Sapphire laser system generated IR pulses with a duration of ~30 fs and optimal center wavelength of 780 nm. The Lund experimental setup allows the tuning of the laser center wavelength from this value up to 794 nm. The pulses are split into two arms, with the most intense part of the beam focused into a gas target filled with argon, to generate an APT centered at a photon energy of about 35 eV and with a spectral envelope of about 12 eV FWHM (full-width-half-maximum). After the attosecond pulse generation, an aluminum filter is used to remove the IR radiation co-propagating with the XUV beam. The second branch of the IR beam is used as a weak probe for the RABBIT technique.

In both experiments, the intensity of the IR probe beam has been measured to be $3 \cdot 10^{11} \text{Wcm}^{-2}$, low enough to ensure the weak field conditions needed for RABBIT. Both arms of the interferometer are actively stabilized in order to minimize sources of systematic errors and ensure stability of the delay in the attosecond range. After recombination of the IR probe with the XUV APT, the two beams further propagate collinearly onto a toroidal mirror that focuses them onto the argon gas jet located inside an electron spectrometer.

In the ETH experiment, the spectrometer contains a reaction microscope detector[51]. This detection scheme allows for the retrieval of the full 3D momentum

vector in coincidence for each individual charged particle over the full $4\pi$ solid angle. In all of the results presented, the data represent the mean value extracted by 15 independent datasets, while the error bars indicate the standard deviation.

In the Lund experiment uses a velocity-map imaging (VMI) detector[52], which measures the projection of the electron distribution onto a position-sensitive detector. This detection technique is well adapted to the geometry of the interaction, with a common XUV and IR polarization axis, chosen to be perpendicular to the detector axis. The 3D-electron momentum distributions are obtained by inversion of the 2D-projections using the pBasex algorithm[53]. The Lund results include 10 datasets, at different fundamental wavelengths.

**Data availability**. The data that support the findings of this study are available from the corresponding author upon reasonable request.

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

## Acknowledgements

C.C., S.H, M.L, L.G and U.K acknowledge support by the ERC advanced Grant No. ERC-2012-ADG_20120216 within the seventh framework program of the European Union and by the NCCR MUST, funded by the Swiss National Science Foundation. A.J.-G. acknowledges support from the DFG QUTIF grant IV 152/6-1. L.A. acknowledges support from the TAMOP NSF Grant No. 1607588, as well as UCF fundings. The Lund group acknowledges support from the ERC advanced grant PALP (339253), the Knut and Alice Wallenberg Foundation, the Swedish Research Council and the Swedish Foundation for Strategic Research. E.L. acknowledges support from the Swedish Research Council, Grant No. 2016-03789. J.M.D. acknowledges support from the Swedish Research Council, Grant No. 2014-03724. C.M., C.L.M.P., A.J.-G., L.A. and F.M. acknowledge computer time from the CCC-UAM and Marenostrum Supercomputer Centers and financial support from the European Research Council under the European Union's Seventh Framework Programme (FP7/2007-2013)/ERC advanced Grant 290853 XCHEM, the MINECO projects FIS2013-42002- R and FIS2016-77889-R, and the European COST Action XLIC CM1204.

## Author contributions

C.C., S.H., and M.L. performed the ETH experiment and analyzed the data. S.Z., D.B., M.I., S.N., S.M., M.G. performed the Lund experiment and analyzed the data. L.R., P.J. built the VMI spectrometer. All the authors were involved in the data interpretation. C. M., C. L. M. P., Á.J.G., L.A., J.M.D., E.L. and F.M. developed the theoretical model. C.C., F.M., A.L. and U.K. wrote the manuscript, which all the authors discussed.

## Additional information

**Competing interests:** The authors declare no competing financial interests.

