## [Peer Review File · Nature Communications]

Reviewers' comments:

Reviewer #1 (Remarks to the Author):

This is an excellent manuscript describing work that deserves publication in Nature Communications. It details the fact that the angular dependence photoionization time delay changes dramatically in the neighborhood of autoionizing resonances, at least in the case of the $3s \rightarrow 5p$ resonance in the argon atom. This is done experimentally and the authors present a theoretical model which exhibits disagreement with the experiment when the autoionization process is omitted. There is one thing, however, that I wish the authors would address. Can they give us some insight into the physics of why the angular distribution of time delay should change so dramatically in the presence of resonances? I think this would add considerably to the paper.

Reviewer #2 (Remarks to the Author):

The present paper investigates ionization time delays in the presence of an autoionizing resonance in argon using the well-known RABBITT technique. Autoionization is a multi-electron effect, which is of considerable interest to many chemists and physicists. Ionization time delays are further of considerable interest to the attosecond community. The present experiment goes beyond conventional RABBITT by recording not only the kinetic energy but also the emission angle of photoelectrons. This refined technique was recently pioneered by the authors (ref. 35) and gives access to the "photoemission time delay anisotropy". The central claim of the present work is that the photoemission time delay anisotropy is sensitive to the presence of an autoionizing resonance. Especially at large angles, a substantial impact of the resonance on the ionization time delay is observed. The claim is substantiated by comparison to appropriate calculations.

While the presented data and conclusions are nice, I am not convinced that they represent a substantial advance over recent work by L'Huillier and co-workers (Ref. [42]) that would justify publication in Nature Communications. Indeed, Ref. [42] is closely related to the present work: the physical system under study is identical; the theoretical results (provided from the same theory group) are obtained using the same model (provided in Ref. [41]); and the experiments are similar in making use of the RABBITT technique. The difference in the experimental techniques is that the present work employs angularly-resolved electron detection, while Ref. [42] used a tunable laser source for their experiments. This allowed L'Huillier and co-workers to scan the XUV photon energy across an autoionizing resonance of Ar ($3s \rightarrow 4p$ instead of $3s \rightarrow 5p$), and record the ionization time delay as a function of the photon energy. Hence, they were able to follow the evolution of the time delay across the resonance. This appears to provide more information on the underlying multi-electron dynamics than the present work. I challenge the present authors to make clear what additional information on the electron dynamics can be extracted from the angularly resolved data?

Generally, I enjoyed reading the paper, as it is well written and interesting. Besides my main concern stated above, I have a few further questions and suggestions to improve the manuscript.

1. Why does the additional time delay caused by the resonance predominantly show up at large angles?
2. Why is the model result without resonance so much shallower for SB 16 as compared to SB 14? Indeed, the effect of the resonance is only dramatic if I compare the experimental data to the model without resonance. If I compare the experimental data for SB 16 to SB 14, the effect is merely larger than the error bars. Thus, speaking of strong correlation effects in the abstract and elsewhere appears

as an overstatement.

3. Under the present conditions, the contribution of the resonance to the emission delay is on the order of 20as when comparing SB 14 and SB 16 data. Would the effect be stronger with a smaller bandwidth? Moreover, how is SB 18 affected by the presence of the resonance at H17?

4. With regard to the sentence "In order to further investigate whether the small effect observed in the β_2 values is due to the presence of the $3s^23p^6 \rightarrow 3s13p65p$ autoionizing resonance, we use the RABBITT method to extract the phases and the associated delays of the ionizing wavepacket.", it is not clear to me how RABBITT can help answering this specific question.

5. The abstract contains some redundancy. I suggest deleting "The atomic delay ... presence of resonances."

6. The term "above threshold ionization" usually refers to the strong-field effect. Thus, it is used incorrectly (or at least confusingly) in "With an IR dressing field present in the RABBITT technique we obtain two-color two-photon above threshold ionization with a photoelectron spectrum that exhibits additional sidebands (SBS) at energies in-between two consecutive APT comb peaks".

7. Improve the clarity of the sentence "These energies correspond to the absorption of an XUV photon together with the additional absorption or emission of the IR photon giving access to the phase difference between the two different ionization channels."

I believe it is confusing for readers not familiar with RABBITT

8. Replace "as" by "of" in "function as the emission angle θ ."

9. In Fig. 4, labels for (a) and (b) are missing. Use consistent line styles in Figs (a) and (b) for theoretical data with and without resonance.

Reviewer #3 (Remarks to the Author):

Cirelli et. al. present angle-resolved RABBITT measurements from argon – essentially angle-resolved photoelectron measurements as a function of energy and XUV-IR time-delay, following ionization via (weak) XUV and IR fields. The results are analysed and discussed first phenomenologically, in terms of the associated characteristics and β - parameters of the distributions, then more precisely in terms of the (angle-resolved) photoionization delays determined by analysis of the RABBITT spectrograms, and comparison with ab initio calculations. This leads to the conclusion that autoionization plays a significant role in some cases, and (through the use of ab initio calculations) the authors are able to verify the contribution of an autoionization process.

The work itself is of a high technical standard, and both the experiments and calculations are challenging. To my knowledge it is one of only a couple of works demonstrating angle-resolved RABBITT, which has only recently become possible (e.g. ref. 35, from many of the current authors), hence experimental determination of angle-resolved photoionization delays. The presentation, however, feels rather perfunctory, with a lot of introduction and not much detailed content (although this is, of course, a subjective view), despite the (objective) quality of the results. This is compounded by rather a lot of

over-selling of what is, at root, (relatively) well-known photoionization physics, albeit studied in a novel manner. Consequently, I've found the work rather difficult to review, although I do think that it is (subjectively) stylistically in line with other high-impact publications in the atto sphere. In the broader context, I think this work is suitable for Nat. Comms. despite the many issues I have with the presentation, although I do think that the manuscript requires significant improvements before being publication ready. In an effort to provide useful feedback to the authors and editors, I have worked through the Nat. Comms. guidelines below, and also made some additional comments.

Questions as per Nat. Comms. Guide

What are the major claims of the paper?

AR-RABBITT measurements of argon form the core experimental results; analysis of these measurements to determine angle-resolved photoionization time-delays and comparison with theory provide the main thrust of the work, and lead to the conclusion that autoionization plays a significant role in some cases.

Are the claims novel? If not, please identify the major papers that compromise novelty.

The experimental method – angle-resolved RABBITT – is fairly challenging and only recently demonstrated (ref. 35, and also partially in Laurent, G., Cao, W., Li, H., Wang, Z., Ben-Itzhak, I., & Cocke, C. L. (2012). Attosecond Control of Orbital Parity Mix Interferences and the Relative Phase of Even and Odd Harmonics in an Attosecond Pulse Train. *Physical Review Letters*, 109(8), 83001.

<https://doi.org/10.1103/PhysRevLett.109.083001>), so could be considered novel. The AR-RABBITT measurements of argon are, to my knowledge, new results.

The novelty in the physics is debateable, and rather more subjective – the core concepts (photoionization, continuum phase effects, autoionization) are not new, but the demonstration in the AR-RABBITT context is novel (although [Swoboda et. al. Phase measurement of resonant two-photon ionization in helium. *Physical Review Letters*, 104, 103003 (2010).

<https://doi.org/10.1103/PhysRevLett.104.103003>] considered the related case of a bound-state resonance), as is the specific application to argon (although ref. 15 presented angle-integrated RABBITT).

For example, the following refs. (and refs. therein) discuss and review these type of effects in “traditional” energy-domain photoelectron spectroscopy – and two references barely scratches the surface of this topic, which has been studied extensively:

- Seaton, M. J. (1983). Quantum defect theory. *Reports on Progress in Physics*, 46(2), 167–257.

<https://doi.org/10.1088/0034-4885/46/2/002>

- Pratt, S. T. (1995). Excited-state molecular photoionization dynamics. *Reports on Progress in Physics*, 58(8), 821–883. <https://doi.org/10.1088/0034-4885/58/8/001>

Will the paper be of interest to others in the field?

I would anticipate that this paper will be of interest to others in the atto-second community.

Will the paper influence thinking in the field?

I suspect not significantly, for the reasons given above: i.e. it is an excellent experimental demonstration, but the underlying physics is (or should be) well-known. Additional discussion on the aspects of the physics that AR-RABBITT measurements might reveal is one area in which the paper might potentially be improved to motivate future thinking (e.g. what aspects of the ionization physics can be measured here, but not in the traditional ways?), since this is where there is actually the potential for new advances in measurement.

Are the claims convincing? If not, what further evidence is needed?

The results, analysis and ab initio calculations are convincing, as is the conclusion that resonant effects play a significant role. The only glaring issue here is a lack of uncertainties on the extracted β -parameters (fig. 2), which undermines the comparison with previous energy-domain results somewhat (see comments below).

Are there other experiments that would strengthen the paper further? How much would they improve it, and how difficult are they likely to be?

None required.

Are the claims appropriately discussed in the context of previous literature?

Again – yes and no, depending on subjective opinion! Purely in the atto-second context, and current stylistic mores, it is fine. But, for my tastes, the massive body of relevant work in the energy-domain literature is (wilfully?) overlooked. It is, however, of note that the authors do reference relevant prior synchrotron work (ref. 28), and benchmark their results against the earlier measurements.

If the manuscript is unacceptable in its present form, does the study seem sufficiently promising that the authors should be encouraged to consider a resubmission in the future?

Again, I find this very hard to judge since I do not particularly like the presentation of the current paper, but I am also reticent to recommend that the authors re-write according to my preferences. I do think that an effort to focus the paper (which is currently around 40% introduction by my count), tighten the language in tone and precision throughout, and provide some more depth to the presentation would significantly improve the manuscript however.

Technical Comments

I'm curious to know if the authors observe any changes in the form of the angular distributions as a function of XUV-IR delay (i.e. time-dependent β values), or only intensity variation (breathing modes)? If so, I think this could be another signature of dynamical effects, e.g. interferences between additional paths such as a direct and autoionizing pathway with different temporal responses. It may be that the temporal resolution and/or statistics are not sufficient to see such effects in the present case.

What IR intensity was used in these experiments? I didn't see it mentioned anywhere, but it is an important parameter since (a) a weak field is implicitly assumed in this work and (b) it is another means of probing dynamical effects.

In fig. 4(a) it appears that there are significant oscillations in the time-delays at small angle, and in fig. 4(b) deviations from calcs. at large angle. These points don't seem to be discussed in the article. Are these results significant and/or interesting?

Further Comments

Introduction and situating of the work in the atto-second context: as mentioned above, the work is placed very specifically in the experimental atto-second context, and the implication seems to be that the core photoionization physics is novel. It is not (at least in general terms); furthermore, the introduction has the feeling of straw man building, since it starts with a discussion of the three-step model of high-harmonic generation. This is (a) a model; (b) a strong-field model; (c) semi-classical. The AR-RABBITT measurements are (a) not strong-field; (b) not in the regime covered by strong-field models; (c) cannot really be described or interpreted by semi-classical models. The authors know this, since they use a fully quantum mechanical scattering treatment for their ab initio calculations, making use of dipole matrix elements and so forth - basically, the well-established machinery of scattering and weak-field photoionization physics. Hence, I think this introductory material is very misleading, particularly to nonexpert readers. Additionally, in my review document, I counted 4 pages of introduction and 6.5 pages of content (excluding figures). This seems rather skewed for a serious scientific publication, where one might expect the content to dominate.

P4 The success of this semi-classical three-step model [4] motivated further investigations in attosecond ionization dynamics to understand how much physical insight can be extracted from this classical picture of the wave packet motion. Based on this picture, the concept of photoemission time delays, first introduced in [5], has been used in the literature [6, 7] to obtain information about the ionization mechanism.

As per the comment above, this is a really, really strange way to introduce the topic. Additionally, I have to question whether ref. 5, from 2010, really "first introduced" the concept of photoemission time delays. Is this not what the Wigner delay is? Or, if one insists on a wavepacket/time-domain picture, is it not already covered by wavepacket treatments of scattering theory (e.g. Rodberg, L. S., & Thaler, R. M. (1967). Introduction to the Quantum Theory of Scattering. Academic Press.; the wideranging review of de Carvalho, C. A. A., & Nussenzveig, H. M. (2002). Time delay. Physics Reports, 364(2), 83–174. [https://doi.org/10.1016/S0370-1573\(01\)00092-8](https://doi.org/10.1016/S0370-1573(01)00092-8)). I agree that the context in the 2010 work is a little different, but the physics is unchanged.

On a related note, the authors make a lot of hay regarding the delay anisotropy (e.g. p5 "*In this work, we clearly demonstrate how the angular dependence of photoionization time delays is strongly affected by correlation effects associated to the mechanism of autoionization, thus giving access to angle-resolved multi-electron dynamics on the attosecond time scale.*" p11 "*Here, we show that additional information, like electron correlation associated to autoionization, is encoded into the delay anisotropy.*") For the most part this seems, to me, rather specious. Of course these effects show up in the delay and delay anisotropy – they affect the magnitudes and phases of the final states populated, which may result in (a)

an observable effect in the angle-integrated RABBITT results and (b) an observable effect in the angle-resolved measurements. This is simply a restatement of energy-domain photoionization physics in the time-domain: in fact, it would be remarkable if these effects don't show up! What is interesting is the fact that the authors have made technically demanding measurements, and obtained details of these effects. For me, the paper would be much more satisfying (and befitting of a Nat. Comms. article aimed more at an expert audience) if the authors focussed on the specific details and what can be learnt here, which is the import and interest of the work (and where AR-RABBITT as an atto-second photoelectron spectroscopy technique might be distinguished from traditional methods), rather than merely stating that expected effects appear, and feigning surprise.

P10 The angular distributions as shown in Fig. 2 are for electrons from HH17 and SB16. They are different because the one extracted for electrons at the energy position of SB16 (panel b) is based on a two-photon absorption process and, thus, is more skewed. This shape is reflected in a non-vanishing value of β_4 being extracted from the fit (Fig. 2d).

This is an example of the often strangely circular and imprecise language, and a lack of rigorous presentation. What was measured? PADs corresponding to a one-photon and two-photon ionization process, at different energies. Of course these would be expected (a priori) to be different, but being "more skewed" (or, indeed, less skewed) does not follow from this statement – rather it is the observation based on the measurement in this particular case. It is not a general result, and it is purely phenomenological.

P10 (& fig. 2) The excellent agreement of our data for the one-photon process (Fig. 2c, red solid line with red solid dots) with prior synchrotron measurements [28] (Fig. 2c, black dots) validates our experiments.

As mentioned above, the authors' efforts of comparing the current measurements with prior results from angle-resolved photoelectron spectroscopy literature are laudable. This does indeed offer a way to benchmark the current time-resolved measurements and maintain a lineage from previous high-resolution energy-domain work. However, the efforts are undermined somewhat by a lack of error bars on the results – this comparison would be much more robust with inclusion of uncertainties (or, at least, an indication/discussion in the text of expected measurement errors).

On a related note, it seemed to me that the β_4 values shown in the line plots (fig. 2d) did not correlate with the colourmap (fig. 1b) at low energy – since all of the distributions appear to show significant 4-fold character here. However, this might be due to intensity and/or renormalization effects, so could purely be perceptual. Again, an error bar on the β -parameters would provide clarity here. (I also found the switch from electron energy to photon energy between figs. 1 and 2 made it difficult for the reader to compare bands to betas, maybe the harmonic/SB order could be added to fig. 1 to consolidate.)

P10 The larger values of β_2 in the case of two-photon transitions are due to the mixing of larger final angular momenta.

I understand what the authors are getting at here, but this is a(nother) imprecise and potentially misleading statement. The mixing of different final state angular momenta indeed would be expected to produce different PADs in the two cases, but there is again no a priori reason that a larger β^2 would be the result. (See, for one example of an unexpected/unintuitive result, Cooper, J., & Zare, R. N. (1968). Angular Distribution of Photoelectrons. *The Journal of Chemical Physics*, 48(2), 942. <https://doi.org/10.1063/1.1668742>). In this specific case the measurement indicates a larger β^2 in the side-bands, but this is not really due to the mixing of larger angular momenta, or at least one cannot say this can be concluded from the results alone... since it's an interference effect, it's rather correct to say that is the coherent sum over these final state components which determines the PADs – but this is not a simple or trivial statement, nor does it define exactly what one should expect in any given case, although it may place upper and lower bounds on the allowed β values. The authors, of course, know this since the theory is detailed in the SM... so why make lose and misleading statements in the main article?

Reviewer #1 (Remarks to the Author):

This is an excellent manuscript describing work that deserves publication in Nature Communications. It details the fact that the angular dependence photoionization time delay changes dramatically in the neighborhood of autoionizing resonances, at least in the case of the $3s \rightarrow 5p$ resonance in the argon atom. This is done experimentally and the authors present a theoretical model which exhibits disagreement with the experiment when the autoionization process is omitted. There is one thing, however, that I wish the authors would address. Can they give us some insight into the physics of why the angular distribution of time delay should change so dramatically in the presence of resonances? I think this would add considerably to the paper.

We thank the referee for his/her positive comments about our work.

In the presence of an autoionization resonance, the relative contribution of the continuum partial waves s and d reached by one-photon absorption changes dramatically due to the interference effect. This changes the photoelectron angular distribution and leads to an anisotropy of the atomic time delay.

In the new version of the manuscript, we have included a new set of data, obtained at Lund University with a different experimental setup, which allows for the measurements of spectrally *and* angle resolved data in the vicinity of an atomic resonance (the $3s^{-1}4p$ resonance).

We also added a discussion part where we address the physical aspects and insights that can be drawn from our resonant and non-resonant data.

Finally, the manuscript has been rewritten to comply for the Nature Communication template as well as to follow the recommendations by the referees (reduced introduction, results, discussion, methods).

Reviewer #2 (Remarks to the Author):

The present paper investigates ionization time delays in the presence of an autoionizing resonance in argon using the well-known RABBITT technique. Autoionization is a multi-electron effect, which are of considerable interest to many chemists and physicists. Ionization time delays are further of considerable interest to the attosecond community. The present experiment goes beyond conventional RABITT by recording not only the kinetic energy but also the emission angle of photoelectrons. This refined technique was recently pioneered by the authors (ref. 35) and gives access to the “photoemission time delay anisotropy“. The central claim of the present work is that the photoemission time delay anisotropy is sensitive to the presence of an autoionizing resonance. Especially at large angles, a substantial impact

of the resonance on the ionization time delay is observed. The claim is substantiated by comparison to appropriate calculations.

While the presented data and conclusions are nice, I am not convinced that they represent a substantial advance over recent work by L'Huillier and co-workers (Ref. [42]) that would justify publication in Nature Communications. Indeed, Ref. [42] is closely related to the present work: the physical system under study is identical; the theoretical results (provided from the same theory group) are obtained using the same model (provided in Ref. [41]); and the experiments are similar in making use of the RABBITT technique. The difference in the experimental techniques is that the present work employs angularly-resolved electron detection, while Ref. [42] used a tunable laser source for their experiments. This allowed L'Huillier and co-workers to scan the XUV photon energy across an autoionizing resonance of Ar ($3s^14p$ instead of $3s^15p$), and record the ionization time delay as a function of the photon energy. Hence, they were able to follow the evolution of the time delay across the resonance. This appears to provide more information on the underlying multi-electron dynamics than the present work. I challenge the present authors to make clear what additional information on the electron dynamics can be extracted from the angularly resolved data?

We thank the referee for his/her comment. We agree with the referee that our experiment is related to that presented by Kotur et al (now Ref. [26]). In the new version of the manuscript, we have actually included experimental results similar to what is presented in Ref. [26], but with, in addition, angular resolution. These results have been obtained by the Lund group by using a tuneable source (exactly as in Ref. [26]) and a VMIS detector, allowing for angular resolution. We present now spectrally and angle resolved phase/time delay in the vicinity of the $3s^14p$ resonance. The new results are shown in the new figure 6 and discussed in the last section of the paper. These results go substantially beyond those presented in our original submission or in Ref. [26], since they represent the first fully differential measurement of atomic delays performed up to date.

We believe that our angle and spectrally resolved results represent the most complete study to date of resonant photoionization time delays. This allows us to characterize the electron wavepacket not only in the time/frequency domain, but even in momentum space. The angular dependence of the time delay can be related to the varying influence of angular momentum channel across the resonance, as explained in our discussion.

In the new version of the manuscript, we have included a new set of data, a discussion part where we address the physical aspects and insights that can be drawn from our resonant and non-resonant data. Finally, the manuscript has been rewritten to comply for the Nature Communication template as well as to follow the recommendations by the referees (reduced introduction, results, discussion, methods).

Generally, I enjoyed reading the paper, as it is well written and interesting. Besides my main concern stated above, I have a few further questions and suggestions to improve the manuscript.

1. Why does the additional time delay caused by the resonance predominantly show up at large angles?

The new result included in the article, which presents the variation of the time delay in the vicinity of the (stronger) $3s^{-1}4p$ resonance, shows a variation of the time delay at all angles (see Fig. 6b).

2. Why is the model result without resonance so much shallower for SB 16 as compared to SB 14? Indeed, the effect of the resonance is only dramatic if I compare the experimental data to the model without resonance. If I compare the experimental data for SB 16 to SB 14, the effect is merely larger than the error bars. Thus, speaking of strong correlation effects in the abstract and elsewhere appears as an overstatement.

In the absence of resonances, the angular anisotropy becomes shallower with increasing electron kinetic energy. This trend is confirmed by experimental data and calculations presented for the case of the helium atom in our previous work, Heuser et al, PRA 94, 063409 (2016), Ref. [31] of the revised manuscript. The anisotropy can be related to the breakdown of the asymptotic approximation for the IR photon absorption, as discussed in the new version of the manuscript (discussion section). This approximation improves as the electron energy increases.

We agree with the referee that the evidence for the $3s^{-1}5p$ resonance (Fig. 5b of the new manuscript) is inferred by comparison with the theoretical calculation with and without resonance. We have now added a result where the effect of a resonance ($3s^{-1}4p$) is clearer. In addition, references to “strong correlation effect” have been removed from the manuscript.

3. Under the present conditions, the contribution of the resonance to the emission delay is on the order of 20 as when comparing SB 14 and SB 16 data. Would the effect be stronger with a smaller bandwidth?

This is a very interesting question. In the recent work by Gruson et al. (Ref. [27] in the reviewed manuscript), the delay is obtained by a spectrally-resolved technique (rainbow RABBIT) and is not affected by the XUV bandwidth (It is however, affected by spectrometer

and IR probe bandwidth). If the delay is obtained by an energy-integrated measurement, as in the present work (Fig. 5 and Fig. 6), the XUV bandwidth also affects the measurement and smoothens the variation of the time delay.

In the new version of the manuscript, we emphasize at the end of the discussion that a quantitative comparison with the experimental data should include bandwidth effects. Those are included in the calculations presented in Fig. 5.

Moreover, how is SB 18 affected by the presence of the resonance at H17?

Sideband 18 is affected by the resonance in both, 5p and 4p, cases. For the 4p case, where the variation with energy is clearest, we do not obtain a variation as simple as the opposite in sign compared to SB16. We attribute this asymmetry to a possible influence of resonances in the final state in both cases (for SB16, the 4s state, as mentioned in the article, and for SB18, higher lying states, including doubly excited states). We chose not to add the results for the time delay obtained from SB18 in the article, which already includes many results. We include, however, the results for (almost) all sidebands for the asymmetry parameters.

4. With regard to the sentence “In order to further investigate whether the small effect observed in the β_2 values is due to the presence of the $3s^23p^6 \rightarrow 3s^13p^65p$ autoionizing resonance, we use the RABBITT method to extract the phases and the associated delays of the ionizing wavepacket.”, it is not clear to me how RABBITT can help answering this specific question.

We agree with the referee and have removed this sentence from the new manuscript. What we meant is that, generally speaking, phase variation is more sensitive to resonance than amplitude variation.

5. The abstract contains some redundancy. I suggest deleting “The atomic delay ... presence of resonances.”

We thank the referee for his/her suggestion and we deleted the sentence. Based also on the comments of Referee 3 (see below) as well as on the additional results in the new manuscript, we have rewritten the second part of the abstract and introduction that we hope to be clearer now.

6. The term “above threshold ionization” usually refers to the strong-field effect. Thus, it is used incorrectly (or at least confusingly) in “With an IR dressing field present in the

RABBITT technique we obtain two-color two-photon above threshold ionization with a photoelectron spectrum that exhibits additional sidebands (SBs) at energies in-between two consecutive APT comb peaks“.

We agree with the referee that the term “above threshold ionization” is mainly used in the case of strong-field ionization. Even if we believe that it is not incorrect to use it also in the case of single-photon (XUV) ionization, we agree that this term may be confusing. For this reason, we removed this term in the revised manuscript. The sentence now reads:

“When an IR dressing field is added, we obtain two-color two-photon transitions with a photoelectron spectrum that exhibits additional sidebands (SBs) at energies in-between two consecutive APT comb peaks.”

7. Improve the clarity of the sentence “These energies correspond to the absorption of an XUV photon together with the additional absorption or emission of the IR photon giving access to the phase difference between the two different ionization channels.”

I believe it is confusing for readers not familiar with RABBITT

Following the suggestion of the referee, we have re-written this sentence and added a reference to the RABBIT method (Ref. [37] of the revised manuscript). More information about the RABBIT method is also given in the discussion section. The new sentence is:

“These energies correspond to the absorption of an XUV photon combined with the additional absorption or emission of an IR photon. Any SB energy can be reached by two different interfering ionization channels [37]”

8. Replace “as” by “of” in “function as the emission angle θ .”

We thank the referee for spotting out this typo, we replaced “as” with “of” in the sentence.

9. In Fig. 4, labels for (a) and (b) are missing. Use consistent line styles in Figs (a) and (b) for theoretical data with and without resonance.

The process of including the data from the Lund experiment required a revision of figure labelled as Fig. 4 in the first manuscript submission. The new Fig. 5 contains now the proper labels to the panels. We also adapted the line styles for the theoretical data: the simulations without (with) resonances are now are represented with dashed (solid) lines.

Reviewer #3 (Remarks to the Author):

Summary

Cirelli et. al. present angle-resolved RABBITT measurements from argon – essentially angle resolved photoelectron measurements as a function of energy and XUV-IR time-delay, following ionization via (weak) XUV and IR fields. The results are analysed and discussed first phenomenologically, in terms of the associated characteristics and β -parameters of the distributions, then more precisely in terms of the (angle-resolved) photoionization delays determined by analysis of the RABBITT spectrograms, and comparison with ab initio calculations. This leads to the conclusion that autoionization plays a significant role in some cases, and (through the use of ab initio calculations) the authors are able to verify the contribution of an autoionization process.

The work itself is of a high technical standard, and both the experiments and calculations are challenging. To my knowledge it is one of only a couple of works demonstrating angle resolved RABBITT, which has only recently become possible (e.g. ref. 35, from many of the current authors), hence experimental determination of angle-resolved photoionization delays. The presentation, however, feels rather perfunctory, with a lot of introduction and not much detailed content (although this is, of course, a subjective view), despite the (objective) quality of the results. This is compounded by rather a lot of over-selling of what is, at root, (relatively) well-known photoionization physics, albeit studied in a novel manner. Consequently, I've found the work rather difficult to review, although I do think that it is (subjectively) stylistically in line with other high-impact publications in the atto sphere. In the broader context, I think this work is suitable for Nat. Comms. despite the many issues I have with the presentation, although I do think that the manuscript requires significant improvements before being publication ready. In an effort to provide useful feedback to the authors and editors, I have worked through the Nat. Comms. guidelines below, and also made some additional comments.

Questions as per Nat. Comms. Guide

What are the major claims of the paper?

AR-RABBITT measurements of argon form the core experimental results; analysis of these measurements to determine angle-resolved photoionization time-delays and comparison with theory provide the main thrust of the work, and lead to the conclusion that autoionization plays a significant role in some cases.

Are the claims novel? If not, please identify the major papers that compromise novelty.

The experimental method – angle-resolved RABBITT – is fairly challenging and only recently demonstrated (ref. 35, and also partially in Laurent, G., Cao, W., Li, H., Wang, Z., Ben-Itzhak, I., & Cocke, C. L. (2012). Attosecond Control of Orbital Parity Mix Interferences

and the Relative Phase of Even and Odd Harmonics in an Attosecond Pulse Train. *Physical Review Letters*, 109(8), 83001. <https://doi.org/10.1103/PhysRevLett.109.083001>), so could be considered novel. The AR-RABBITT measurements of argon are, to my knowledge, new results.

The novelty in the physics is debateable, and rather more subjective – the core concepts (photoionization, continuum phase effects, autoionization) are not new, but the demonstration in the AR-RABBITT context is novel (although [Swoboda et. al. Phase measurement of resonant two-photon ionization in helium. *Physical Review Letters*, 104, 103003 (2010). <https://doi.org/10.1103/PhysRevLett.104.103003>] considered the related case of a bound-state resonance), as is the specific application to argon (although ref. 15 presented angle-integrated RABBITT).

We thank the referee for these comments. The paper from Swoboda et al. is now included as Ref. [16] in the revised manuscript. Indeed, this pioneering work on resonant two-color two-photon ionization of helium through the $1s3p^1P_1$ state present important similarities with our experiment. We also included the reference to Laurent et al. [29], as well as the recent article by Villeneuve et al. [30], which both use an angular-resolved RABBIT technique.

For example, the following refs. (and refs. therein) discuss and review these type of effects in “traditional” energy-domain photoelectron spectroscopy – and two references barely scratches the surface of this topic, which has been studied extensively:

- Seaton, M. J. (1983). Quantum defect theory. *Reports on Progress in Physics*, 46(2), 167–257. <https://doi.org/10.1088/0034-4885/46/2/002>
- Pratt, S. T. (1995). Excited-state molecular photoionization dynamics. *Reports on Progress in Physics*, 58(8), 821–883. <https://doi.org/10.1088/0034-4885/58/8/001>

We have added these two references in the revised version of the manuscript, they are [21] and [22].

Will the paper be of interest to others in the field?

I would anticipate that this paper will be of interest to others in the atto-second community.

Will the paper influence thinking in the field?

I suspect not significantly, for the reasons given above: i.e. it is an excellent experimental demonstration, but the underlying physics is (or should be) well-known. Additional discussion on the aspects of the physics that AR-RABBITT measurements might reveal is one area in which the paper might potentially be improved to motivate future thinking (e.g. what

aspects of the ionization physics can be measured here, but not in the traditional ways?), since this is where there is actually the potential for new advances in measurement.

We have added an entire discussion section to the revised manuscript where different aspects of angular-RABBIT are being discussed (photoelectron angular distributions, time delays, non-resonant and resonant ionization). We hope it will motivate further thinking about the interest of the technique.

Are the claims convincing? If not, what further evidence is needed?

The results, analysis and ab initio calculations are convincing, as is the conclusion that resonant effects play a significant role. The only glaring issue here is a lack of uncertainties on the extracted β -parameters (fig. 2), which undermines the comparison with previous energy-domain results somewhat (see comments below).

In the revised version of the manuscript, we have included the uncertainties for the beta-parameters, see new Fig. 3. We are also including now results from another experiment, performed in Lund.

Are there other experiments that would strengthen the paper further? How much would they improve it, and how difficult are they likely to be?

None required.

As written above, we have now included new experimental data that had been independently performed at Lund University with a different detection scheme. As shown in the updated Fig. 2, due to a slightly different fundamental wavelength, HH17 in the Lund data is resonant with the $3s^{-1}4p$ autoionizing state, while in the ETH experiment it is resonant with the $3s^{-1}5p$ state. This allows us to present the data even for two distinct resonances.

Moreover, the experimental setup of the Lund experiment allows the energy tuning of HH17 across the 4p resonance, such that angle *and* spectrally resolved atomic delays can be extracted. While the same approach would be in principle possible to be implemented for the 5p resonance, in the spectral analysis we focused on the 4p resonance because the effects induced on the atomic phase are expected to be more significant due to its larger energy width (see the values of Gamma tabulated for instance in Ref. [46]).

Are the claims appropriately discussed in the context of previous literature?

Again – yes and no, depending on subjective opinion! Purely in the atto-second context, and current stylistic mores, it is fine. But, for my tastes, the massive body of relevant work in the

energy-domain literature is (wilfully?) overlooked. It is, however, of note that the authors do reference relevant prior synchrotron work (ref. 28), and benchmark their results against the earlier measurements.

We have added a few references in the new manuscript (without pretending to cover the vast energy-domain research field), they are listed as [24], [25] and also [32], [33].

If the manuscript is unacceptable in its present form, does the study seem sufficiently promising that the authors should be encouraged to consider a resubmission in the future?

Again, I find this very hard to judge since I do not particularly like the presentation of the current paper, but I am also reticent to recommend that the authors re-write according to my preferences. I do think that an effort to focus the paper (which is currently around 40% introduction by my count), tighten the language in tone and precision throughout, and provide some more depth to the presentation would significantly improve the manuscript however.

We followed the suggestions of the referee and reduced the size of the introduction and revised the text throughout all the manuscript. At the same time, we added a complete new section with the discussion of the results, where we present a new in-depth analysis of spectrally and angle resolved data.

Technical Comments

I'm curious to know if the authors observe any changes in the form of the angular distributions as a function of XUV-IR delay (i.e. time-dependent β values), or only intensity variation (breathing modes)? If so, I think this could be another signature of dynamical effects, e.g. interferences between additional paths such as a direct and autoionizing pathway with different temporal responses. It may be that the temporal resolution and/or statistics are not sufficient to see such effects in the present case.

Following the suggestion of the referee, we extracted the values of the β_2 parameter as a function of the time delay. In the newly added Fig. 4, we present delay-dependent β_2 value for SB14, SB16, SB18, SB22 and H17. We can draw several conclusions:

- 1) While for the harmonics the amplitude of β_2 oscillations remains constant over all the investigated spectral range, for the sidebands, it decreases as the photon energy increases, as shown in Fig. 4(b). We attribute this trend to the general fact that atomic delay becomes more and more independent on the electron emission angle as the excitation energy increases.
- 2) There seems to be no trace of resonance effects induced on the amplitude of the sideband oscillations. Indeed, the trend of the curve smoothly decreases with increasing sideband order

even in the case of SB16 (at photon energy of 26.6 eV), which is resonant with the 4p autoionizing state (see also Fig. 1). This result suggests that the amplitude of β_2 is not a good observable to investigate correlation effects induced by autoionizing states, which instead manifest themselves clearly in delay-integrated β_2 curve (Fig. 3d) and even more significantly in the angular resolved atomic delay (Fig. 5 and 6).

In the new discussion session, we comment on the reason for the oscillation of the β_2 parameter, and its decrease as the kinetic energy increases (page 18, just after Eq. 7). This is due to the asymmetry in the absorption and emission paths of the RABBIT interferometer, which decreases as the kinetic energy increases.

What IR intensity was used in these experiments? I didn't see it mentioned anywhere, but it is an important parameter since (a) a weak field is implicitly assumed in this work and (b) it is another means of probing dynamical effects.

We apologize to the referee for not mentioning the intensity of the IR field used in the experiment. Indeed, this is quite an important parameter to determine whether the experiments have been conducted in the weak field assumption needed for RABBIT.

“The intensity of the IR probe beam has been measured to be 3×10^{11} W/cm², low enough to ensure the weak field conditions needed for RABBIT”.

This sentence has been added in the methods section.

In fig. 4(a) it appears that there are significant oscillations in the time-delays at small angle, and in fig. 4(b) deviations from calcs. at large angle. These points don't seem to be discussed in the article. Are these results significant and/or interesting?

We think that the oscillations in the atomic delay for small angles for SB14 are not due to any physical effect, but to the noise in the data, related to the normalization procedure used in the ETH data. We believe so for two reasons:

1) The newly added data, measured with a complete different experimental setup in Lund (employing a VMIS detector instead of a COLTRIMS) present a much smaller experimental uncertainty and do not show the same oscillatory behaviour as the data collected at ETH (Fig. 5a).

2) Furthermore, the theoretical model does not predict any of these features and it would be hard to attribute them to any meaningful effect.

For this reason, we do not comment about them in our manuscript, but we focus our attention on the general trend well reproduced by the model.

Further Comments

Introduction and situating of the work in the atto-second context: as mentioned above, the work is placed very specifically in the experimental atto-second context, and the implication seems to be that the core photoionization physics is novel. It is not (at least in general terms); furthermore, the introduction has the feeling of straw man building, since it starts with a discussion of the three-step model of high-harmonic generation. This is (a) a model; (b) a strong-field model; (c) semi-classical. The AR-RABBITT measurements are (a) not strong-field; (b) not in the regime covered by strong-field models; (c) cannot really be described or interpreted by semi-classical models. The authors know this, since they use a fully quantum mechanical scattering treatment for their ab initio calculations, making use of dipole matrix elements and so forth - basically, the well-established machinery of scattering and weak-field photoionization physics. Hence, I think this introductory material is very misleading, particularly to nonexpert readers. Additionally, in my review document, I counted 4 pages of introduction and 6.5 pages of content (excluding figures). This seems rather skewed for a serious scientific publication, where one might expect the content to dominate.

We thank the referee for his/her comment and the suggestion to review the introduction of our work. We have revised this part of the manuscript, removing the references to the three-step model and the determination of photoionization time delays in the strong-field case. The introduction has been reduced to 2 pages, and a discussion of about 4 pages has been added.

P4 The success of this semi-classical three-step model [4] motivated further investigations in attosecond ionization dynamics to understand how much physical insight can be extracted from this classical picture of the wave packet motion. Based on this picture, the concept of photoemission time delays, first introduced in [5], has been used in the literature [6, 7] to obtain information about the ionization mechanism.

As per the comment above, this is a really, really strange way to introduce the topic. Additionally, I have to question whether ref. 5, from 2010, really “first introduced” the concept of photoemission time delays. Is this not what the Wigner delay is? Or, if one insists on a wavepacket/time-domain picture, is it not already covered by wavepacket treatments of scattering theory (e.g. Rodberg, L. S., & Thaler, R. M. (1967). Introduction to the Quantum Theory of Scattering. Academic Press.; the wideranging review of de Carvalho, C. A. A., & Nussenzveig, H. M. (2002). Time delay. Physics Reports, 364(2), 83–174. [https://doi.org/10.1016/S0370-1573\(01\)00092-8](https://doi.org/10.1016/S0370-1573(01)00092-8)). I agree that the context in the 2010 work is a little different, but the physics is unchanged.

We agree with the referee comment and changed the introduction. We have also changed how the references are presented in the introduction. The Schultze et al paper (now Ref. [2]) is

referenced as the first work which determined experimentally photoionization time delays as introduced theoretically by the original Wigner and Smith papers ([4] and [5]) and reviewed by the de Carvalho and Nussenzweig work ([6]).

On a related note, the authors make a lot of hay regarding the delay anisotropy (e.g. p5 “In this work, we clearly demonstrate how the angular dependence of photoionization time delays is strongly affected by correlation effects associated to the mechanism of autoionization, thus giving access to angle-resolved multi-electron dynamics on the attosecond time scale.” p11 “Here, we show that additional information, like electron correlation associated to autoionization, is encoded into the delay anisotropy.”) For the most part this seems, to me, rather specious. Of course these effects show up in the delay and delay anisotropy – they affect the magnitudes and phases of the final states populated, which may result in (a) an observable effect in the angle-integrated RABBITT results and (b) an observable effect in the angle resolved measurements. This is simply a restatement of energy-domain photoionization physics in the time-domain: in fact, it would be remarkable if these effects don’t show up! What is interesting is the fact that the authors have made technically demanding measurements, and obtained details of these effects. For me, the paper would be much more satisfying (and befitting of a Nat. Comms. article aimed more at an expert audience) if the authors focussed on the specific details and what can be learnt here, which is the import and interest of the work (and where AR-RABBITT as an atto-second photoelectron spectroscopy technique might be distinguished from traditional methods), rather than merely stating that expected effects appear, and feigning surprise.

In the revised version of the manuscript we present a new section where we show angle and spectrally resolved data for the 4p resonance. The new spectral analysis for the 4p resonance shows that for resonances with a large width, the experimental energy resolution is enough to observe the expected (see Fig. 7) large variation of the atomic delay as a function of the excitation energy.

The manuscript is also including now a discussion of what can be learnt (or not) in an angular-resolved RABBIT measurement.

P10 The angular distributions as shown in Fig. 2 are for electrons from HH17 and SB16. They are different because the one extracted for electrons at the energy position of SB16 (panel b) is based on a two-photon absorption process and, thus, is more skewed. This shape is reflected in a non-vanishing value of β_4 being extracted from the fit (Fig. 2d).

This is an example of the often strangely circular and imprecise language, and a lack of rigorous presentation. What was measured? PADs corresponding to a one-photon and two-photon ionization process, at different energies. Of course these would be expected (a priori)

to be different, but being “more skewed” (or, indeed, less skewed) does not follow from this statement – rather it is the observation based on the measurement in this particular case. It is not a general result, and it is purely phenomenological.

We agree with the referee that the sentence that he/she highlighted in this comment was not bringing any important information to the readers. Therefore, while leaving the plots of the PADs for HH17 and SB16, we removed the comments on them and focused the discussion on the interesting curves for the extracted β_2 parameters.

P10 (& fig. 2) *The excellent agreement of our data for the one-photon process (Fig. 2c, red solid line with red solid dots) with prior synchrotron measurements [28] (Fig. 2c, black dots) validates our experiments.*

As mentioned above, the authors’ efforts of comparing the current measurements with prior results from angle-resolved photoelectron spectroscopy literature are laudable. This does indeed offer a way to benchmark the current time-resolved measurements and maintain a lineage from previous highresolution energy-domain work. However, the efforts are undermined somewhat by a lack of error bars on the results – this comparison would be much more robust with inclusion of uncertainties (or, at least, an indication/discussion in the text of expected measurement errors).

We have now added in Fig. 3 (that replaces Fig. 2 of the first submission) the error bars as they are retrieved from the fit. We performed the same analysis on the delay-integrated momentum distributions for all the 15 individual data sets and calculated for β_2 and β_4 the mean value. The error bars are determined by calculating the standard deviation of the 15 values extracted by the individual data sets.

On a related note, it seemed to me that the β_4 values shown in the line plots (fig. 2d) did not correlate with the color map (fig. 1b) at low energy – since all of the distributions appear to show significant 4-fold character here. However, this might be due to intensity and/or renormalization effects, so could purely be perceptual. Again, an error bar on the β -parameters would provide clarity here. (I also found the switch from electron energy to photon energy between figs. 1 and 2 made it difficult for the reader to compare bands to betas, maybe the harmonic/SB order could be added to fig. 1 to consolidate.)

The values of the β_4 parameters are extracted from the fit of the photoelectron angular distributions with the target equation written in the text only for energies corresponding to the sideband signals, where 2-photon transitions are involved.

In the revised manuscript we have added the data measured in a separate experiment with a different experimental apparatus. The fact that the value of β_4 (and β_2) independently retrieved in both experiments are in a fairly good quantitative agreement (Fig. 3c,d) allows us to conclude that the extracted values are correct.

As suggested by the referee, we changed the representation of the data in Fig. 1, which are now shown as a function of the excitation photon energy. In this way, a better comparison with the curves shown in all the other figures is more straightforward.

P10 The larger values of β_2 in the case of two-photon transitions are due to the mixing of larger final angular momenta.

I understand what the authors are getting at here, but this is a(nother) imprecise and potentially misleading statement. The mixing of different final state angular momenta indeed would be expected to produce different PADs in the two cases, but there is again no a priori reason that a larger β_2 would be the result. (See, for one example of an unexpected/unintuitive result, Cooper, J., & Zare, R. N. (1968). Angular Distribution of Photoelectrons. *The Journal of Chemical Physics*, 48(2), 942. <https://doi.org/10.1063/1.1668742>). In this specific case the measurement indicates a larger β_2 in the side-bands, but this is not really due to the mixing of larger angular momenta, or at least one cannot say this can be concluded from the results alone... since it's an interference effect, it's rather correct to say that is the coherent sum over these final state components which determines the PADs – but this is not a simple or trivial statement, nor does it define exactly what one should expect in any given case, although it may place upper and lower bounds on the allowed β values. The authors, of course, know this since the theory is detailed in the SM... so why make lose and misleading statements in the main article?

We agree with the referee that our statement may sound misleading and imprecise. Therefore, following the suggestion of the referee, we have removed it from the revised version of the manuscript.

REVIEWERS' COMMENTS:

Reviewer #1 (Remarks to the Author):

This is a revised manuscript; revised so extensively that even the title has changed. The authors have responded to every one of the referees' comment and produced, in my view, a much better paper. Many ambiguous or misleading statements have been removed or revised, and much more physical insight is provided. In a general sense, the paper is a review of the physics of the energy and angular distribution of attosecond time delay in the vicinity of a resonance so it is quite general.

I now strongly recommend publication as is.

Reviewer #2 (Remarks to the Author):

I would like to thank the authors for the thorough revision and the new data. The combination of the two techniques (angle resolved RABBIT, and tuning the laser frequency), and the new theory figure, add great value to the manuscript. This comprehensive work now deserves publication in Nature Communications.

All questions from the previous review were answered adequately. I only have a couple of remarks.

1. Check references to equations. Equation 2 (the Legendre polynomials) is referenced as eq. 1 at least twice.

2. The authors speak of a low detector efficiency at $\theta = 0$. Isn't the reason for the minimum at $\theta = 0$ merely a geometrical effect related to the solid angle? Indeed, the fit function (equation 2) is multiplied with a factor of $\sin(\theta)$ when applied to the experimental data.

3. On page 17, last paragraph it is probably better to speak of the photoelectron angular distribution rather than "the emission angle".

4. In the sentence right after, I would speak of different pathways leading to the same ionization channel (i.e. SB 16), not different ionization channels (e.g. SB16, HH17, SB 18, ...). In the notation used to refer to the pathways, I would suggest to either leave out the principal quantum number, or state it in each step.

Reviewer #3 (Remarks to the Author):

Firstly, I must say that the authors deserve credit for their careful and thorough work addressing the comments of the referees. The resubmitted work contains significant changes to the manuscript which, in my opinion, make for a much clearer, and stronger, presentation of the work. The addition of new experimental results (and authors) adds further depth. I think the manuscript, in its current form, is suitable for publication in Nat. Comms.

In terms of the science, I only have a couple of additional comments, which I include here simply for the sake of discussion and completeness – no further manuscript changes are required unless the authors feel so inclined.

The data presented in fig. 3(c) & (d) seem to show quite a bit of disagreement between the ETH and Lund results, while the text asserts that they “compare well with the synchrotron data”. Do the authors have a further comment here? It seemed to me that the beta parameters away from the resonance region should match here within experimental error, but maybe I missed something.

The discussion about fig. 4 (p18) suggests that the loss of modulation depth is due to the relative amplitudes of the channels. However, as detailed in ref. [A], it seems just as likely that this is due to changes in the partial wave phases. This would be consistent with the phase shift, and suggestion of an asymmetric temporal profile, for SB22 in fig. 4. Perhaps the authors could comment further based on their full numerical results, rather than just interpret the results based on the simplified model presented in the discussion section?

[A] Hockett, P. (2017). Angle-resolved RABBITT: theory and numerics. *Journal of Physics B: Atomic, Molecular and Optical Physics*, 50(15), 154002. <http://doi.org/10.1088/1361-6455/aa7887>

Reviewer #1 (Remarks to the Author):

This is a revised manuscript; revised so extensively that even the title has changed. The authors have responded to every one the the referees' comment and produced, in my view, a much better paper. Many ambiguous or misleading statements have been removed or revised, and much more physical insight is provided. In an general sense, the paper is review of the physics of the energy and angular distribution of attosecond time delay in the vicinity of a resonances so it is quite general.

I now strongly recommend publication as is.

We thank the referee for his/her positive feedback and we are glad to receive the recommendation to publish our manuscript in the present form.

Reviewer #2 (Remarks to the Author):

I would like to thank the authors for the thorough revision and the new data. The combination of the two techniques (angle resolved RABBITT, and tuning the laser frequency), and the new theory figure, add great value to the manuscript. This comprehensive work now deserves publication in Nature Communications.

All questions from the previous review were answered adequately. I only have a couple of remarks.

1. Check references to equations. Equation 2 (the Legendre polynomials) is referenced as eq. 1 at least twice.

We thank the referee for finding these typos; we have changed the wrong references at the bottom of page 9. Now they are both pointing correctly towards Eq. (2).

2. The authors speak of a low detector efficiency at $\theta = 0$. Isn't the reason for the minimum at $\theta = 0$ merely a geometrical effect related to the solid angle?

Indeed, the fit function (equation 2) is multiplied with a factor of $\sin(\theta)$ when applied to the experimental data.

The referee is right. What we mean with the statement of "low detection efficiency at $\theta = 0$ " is exactly the effect due to the solid angle. In our experiments, the COLTRIMS detector indeed directly records differential photoionization cross-sections and inherently it is "not efficient" to detect charged particle along the $\theta = 0$ direction, due to the need of multiplying the momentum distributions by the $\sin(\theta)$ factor.

Following the suggestion of the referee, we have changed the sentence into:

"The green lines represent the fit of the distributions, which account for the detector geometry [Eq. (2) is multiplied by $\sin(\theta)$ to account for the geometrical effect related to the solid angle]."

3. On page 17, last paragraph it is probably better to speak of the photoelectron angular distribution rather than “the emission angle” .

We agree with the referee and changed the sentence according to his/her suggestion.

4. In the sentence right after, I would speak of different pathways leading to the same ionization channel (i.e. SB 16), not different ionization channels (e.g. SB16, HH17, SB 18, ...).
In the notation used to refer to the pathways, I would suggest to either leave out the principal quantum number, or state it in each step.

Following the suggestion of the referee, we have changed the sentence, which now reads:
“For each ionization channel leading to the same final state, our experimental measurement involves three two-photon ionization pathways: $3p \rightarrow \lambda \rightarrow \ell$, with $(\lambda, \ell) = (0, 1), (2, 1), (2, 3)$.”

Reviewer #3 (Remarks to the Author):

Firstly, I must say that the authors deserve credit for their careful and thorough work addressing the comments of the referees. The resubmitted work contains significant changes to the manuscript which, in my opinion, make for a much clearer, and stronger, presentation of the work. The addition of new experimental results (and authors) adds further depth. I think the manuscript, in its current form, is suitable for publication in Nat. Comms.

In terms of the science, I only have a couple of additional comments, which I include here simply for the sake of discussion and completeness – no further manuscript changes are required unless the authors feel so inclined.

The data presented in fig. 3(c) & (d) seem to show quite a bit of disagreement between the ETH and Lund results, while the text asserts that they “compare well with the synchrotron data”. Do the authors have a further comment here? It seemed to me that the beta parameters away from the resonance region should match here within experimental error, but maybe I missed something.

We agree with the referee that the two experimental datasets presented in Fig. 3 should match within the experimental error. We do not have a clear explanation while they do not.

We can speculate that the different excitation spectra (shown in Fig. 1) may play a role. Indeed, the ETH spectrum (red line in Fig. 1) has narrower harmonics with respect to the ones measured in Lund (blue line in Fig. 1). This may explain the better agreement of the ETH data with synchrotron measurements (Fig. 3c).

On the other hand additional effects, like for instance those induced by the calibration of the different momentum image detectors (COLTRIMS and VMIS) can be important.

While it is undoubtedly interesting to investigate this issue further, we still believe that the common trend shown in Fig. 3c and 3d for both datasets (and the fair agreement with the synchrotron data) is significant enough to validate the experiments.

The discussion about fig. 4 (p18) suggests that the loss of modulation depth is due to the relative amplitudes of the channels. However, as detailed in ref. [A], it seems just as likely that this is due to changes in the partial wave phases. This would be consistent with the phase shift, and suggestion of an asymmetric temporal profile, for SB22 in fig. 4. Perhaps the authors could comment further based on their full numerical results, rather than just interpret the results based on the simplified model presented in the discussion section?

[A] Hockett, P. (2017). Angle-resolved RABBITT: theory and numerics. *Journal of Physics B: Atomic, Molecular and Optical Physics*, 50(15), 154002.
<http://doi.org/10.1088/1361-6455/aa7887>

We thank the referee for this comment and for the reference to the manuscript by P. Hockett. Indeed, the change in the beta modulations might be due to changes in the partial wave phases. As noted by the referee, SB22 does exhibit a slight phase shift, which we did not comment in the manuscript (and which we did not attempt to reproduce in our theoretical calculations).

We added the following sentence in the text:

“Alternatively, as suggested in [49], this behavior might also result from changes in the phases involved in the two-photon transitions as the photon energy increases.”